# The architecture of transmembrane and cytoplasmic juxtamembrane regions of Toll-like receptors

F. D. Kornilov[1,2,6], A. V. Shabalkina [1,2,6], Cong Lin[3,6], P. E. Volynsky[1,4], E. F. Kot [1,2], A. L. Kayushin[1], V. A. Lushpa [1,2], M. V. Goncharuk[1], A. S. Arseniev[1], S. A. Goncharuk [1,2,7] ✉, Xiaohui Wang [3,5,7] ✉ & K. S. Mineev [1,2,7] ✉

Toll-like receptors (TLRs) are the important participants of the innate immune response. Their spatial organization is well studied for the ligand-binding domains, while a lot of questions remain unanswered for the membrane and cytoplasmic regions of the proteins. Here we use solution NMR spectroscopy and computer simulations to investigate the spatial structures of transmembrane and cytoplasmic juxtamembrane regions of TLR2, TLR3, TLR5, and TLR9. According to our data, all the proteins reveal the presence of a previously unreported structural element, the cytoplasmic hydrophobic juxtamembrane α-helix. As indicated by the functional tests in living cells and bioinformatic analysis, this helix is important for receptor activation and plays a role, more complicated than a linker, connecting the transmembrane and cytoplasmic parts of the proteins.

Toll-like receptors (TLRs) are important members of the innate immunity system. By recognizing the pathogen-associated molecular patterns, these proteins launch the inflammation[1]. They are involved in a variety of severe diseases including cancers and therefore are considered prospective targets for novel therapies[2,3]. In this regard, TLRs attract much interest in structural biology. TLRs are typical representatives of single-pass membrane proteins. They are composed of a single transmembrane α-helix that connects the massive 60–90 kDa ligand-binding domain at the N-terminus and the 20 kDa cytoplasmic Toll-Interleukin-1 receptor homology (TIR) domain at the C-terminus[4]. All TLRs are active as homo- or heterodimers[5]. Human TLRs can be divided into five subfamilies, according to the phylogeny[6]. TLR3, TLR4, and TLR5 form their own subfamilies, with only one member; TLRs 1, 2, and 6 all recognize the lipopeptides, and together with TLR10 with no identified ligand they form the largest subfamily; and TLRs 7, 8, and 9

form the second-largest subfamily of TLRs, capable of the nucleic acids recognition[7].

A complete understanding of the TLR spatial organization still has to be achieved. Structures of the ligand-binding domains were solved by X-ray for all human TLRs except TLR10[8–16]; however, the TIR domain structures were reported only for the representatives of the TLR1 subfamily[17–21], and the major "blank spot" in the structure of TLRs is the conformation of their transmembrane domains (TMDs) and cytoplasmic juxtamembrane (JM) regions. Both parts are believed to play an important role in TLR activation. In particular, isolated TMDs of all TLRs are capable of dimerization[22] and transmembrane peptides, corresponding to the TMDs of TLR2, were shown to inhibit the full-length receptors[23]. Moreover, a polymorphism was found in the TMD of TLR1, which modulates the immune response and is associated with Crohn's disease[24]. The role of the cytoplasmic juxtamembrane domain

[1]Shemyakin-Ovchinnikov Institute of Bioorganic Chemistry, Moscow 117997, Russia. [2]Moscow Institute of Physics and Technology, Dolgoprudny 141701, Russia. [3]Laboratory of Chemical Biology, Changchun Institute of Applied Chemistry, Chinese Academy of Sciences, 130022 Changchun, Jilin, China. [4]Institute of Cytology of Russian Academy of Sciences, Tikhoretsky 4, 194064 Saint Petersburg, Russia. [5]School of Applied Chemistry and Engineering, University of Science and Technology of China, 230026 Hefei, Anhui, China. [6]These authors contributed equally to the work: F.D. Kornilov, A.V. Shabalkina, Cong Lin. [7]These authors jointly supervised this work: S.A. Goncharuk, Xiaohui Wang, K.S. Mineev. ✉e-mail: ms.goncharuk@gmail.com; xiaohui.wang@ciac.ac.cn; mineev@nmr.ru

was demonstrated for the TLR4 receptor—deletion of this region, as well as alterations in its primary structure, were shown to alter substantially the ability of the receptor to be activated by LPS and to oligomerize[25,26]. Despite the obvious importance, the structure of both the TMD and JM regions seems poorly investigated. There are three cryo-EM structures of full-length TLRs: TLR5 homodimer[27] and TLR3 and TLR7 in complex with the signaling regulator UNC93B1[28]. However, the first one was solved with a resolution as low as 2.6 nm which precluded the analysis of TMDs, and in the TLR3 and TLR7 structures, the density of the cytoplasmic TIR domains and JM regions was not observed. Our group investigated the structures of TMDs of two TLR members − TLR3 and TLR4 by NMR in solution, the latter was taken with potential JM regions[29,30]. According to our previous data, TLR4 TMD is a long and straight helix comprising both the predicted TMD and JM regions[30]. The most primitive analysis of the amino acid sequences of other TLRs suggests that a similar architecture of TM domains may be observed: all the proteins contain the highly hydrophobic "JM" regions that could be parts of their TMDs[30].

Here we study four TLR TMDs from different subfamilies using NMR in solution and taking the most biologically relevant membrane mimetic and ambient conditions. Combined with functional studies and bioinformatic analysis, we show that JM regions are important for receptor activation and play a role, more complicated than a TMD-TIR linker.

## Results

### Selection of objects and membrane mimetic environment

As a first step of the study, we analyzed the hydrophobic properties of TMDs and adjacent cytoplasmic regions of all human TLRs, the result is shown in Fig. 1. In order to properly locate the possible juxtamembrane domains, we used the moving average hydrophobicity with the frame of four residues, which is close to the step of an α-helix. As we have found out, all TLRs are very similar in specified regions, with the single exception of TLR4. TMDs are connected to the globular TIR domains with the linkers that have a length of 28–32 residues, and 14–18 juxtamembrane residues are hydrophobic, suggesting the formation of an additional membrane-associated helix. In the case of TLR4, the linker region is much shorter and contains only 19 amino acids, the juxtamembrane region is highly hydrophobic and short. No separation between the transmembrane and juxtamembrane parts is observed, which is in agreement with our previously reported NMR structure of TLR4[30].

Lengths of the hydrophobic juxtamembrane patches are similar within the subfamilies − TLR1/2/6/10 and TLR7/8/9. JM parts of TLR3

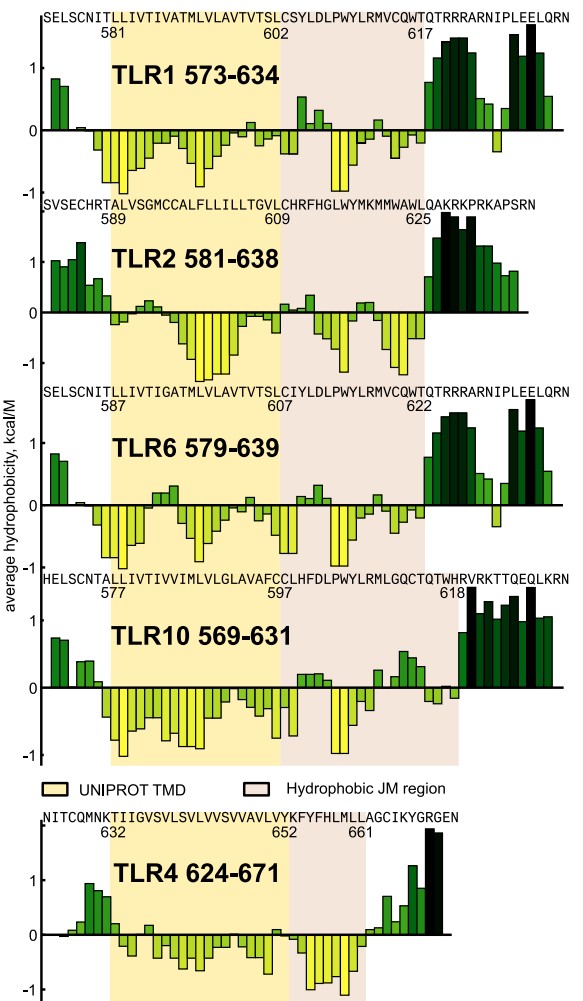
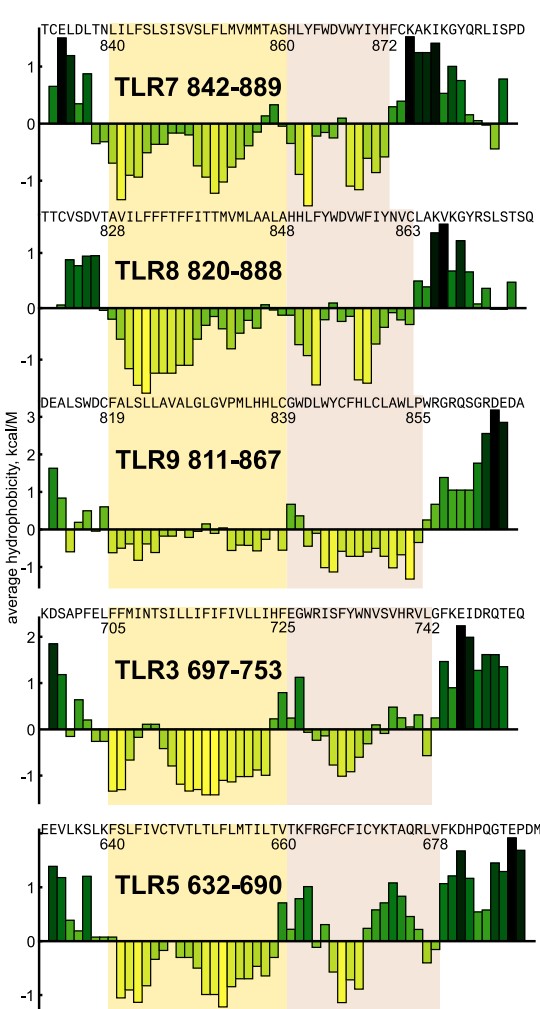

**Fig. 1 | Hydrophobic properties of TLR juxtamembrane residues.** The moving average hydrophobicity (the frame equals four amino acids) is shown for the TMDs and JM parts of all human TLRs. Bars are colored with respect to the hydrophobicity values: yellow for hydrophobic, and dark green for hydrophilic. All sequences are taken starting from the 8th residue prior to the predicted TMDs and finishing with the first residue of the presumed TIR domain and are aligned by the position of the TMD. The location of TMDs is taken according to the UNIPROT database and is highlighted by a yellow background. Hydrophobic juxtamembrane regions are highlighted by the light-red background. Panels are grouped with respect to the TLR subfamilies. Source data are provided as a Source Data file.

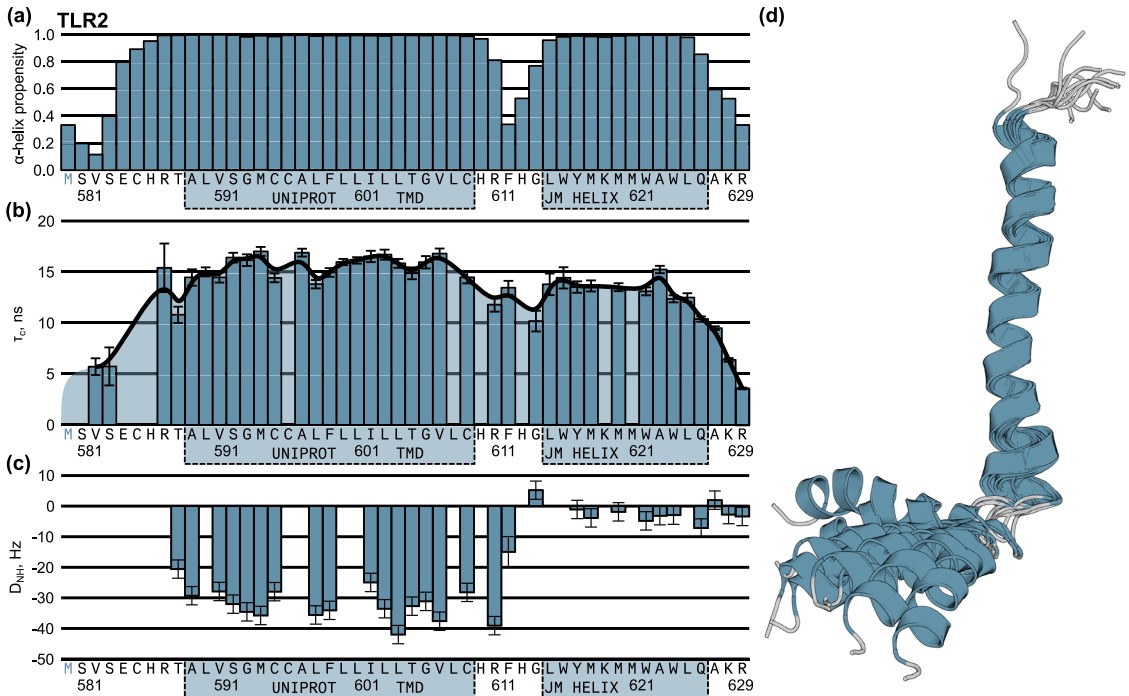

**Fig. 2 | Spatial structure of TLR2tmjm in DMPC/DMPG/DHPC bicelles (DMPC:DMPG = 4:1). a** The probability of α-helix conformation of the amino acid residues, according to the chemical shift analysis. **b** Values of $\tau_C$, correlation time of rotational diffusion, for the individual amide groups of the amino acid residues. **c** Magnitudes of residual dipolar couplings ($^1D_{NH}$). The sequence is numbered according to UNIPROT. The blue boxes denote the parts of TLR2tmjm: the region that is annotated in UNIPROT as the transmembrane domain and the region that forms the juxtamembrane α-helix. The blue color of the first Met indicates that it is not included in the original sequence of TLR2. **d** The ten best NMR structures are superimposed on the backbone atoms of transmembrane α-helix. Source data are provided as a Source Data file.

and TLR5 are relatively polar and contain only the short patches of hydrophobic residues, which are separated from the predicted TM domains by 4–5 polar amino acids. Thus, we assumed that it is sufficient to investigate the structure of one representative of each TLR subfamily to get a complete understanding of their TM and JM domain structural organization. In this regard, we engineered four homologous protein constructs, containing 7–8 extracellular residues, TMD, and 20–21 adjacent cytoplasmic residues of TLR2 (TLR2tmjm), TLR3 (TLR3tmjm), TLR5 (TLR5tmjm), and TLR9 (TLR9tmjm). Together with TLR4, studied previously, these constructs constitute the set of TLR TM domains from all the five subfamilies.

All four proteins were synthesized using the *E.coli*-based cell-free system, purified, and their NMR spectra were recorded in several membrane mimetics: DPC and LPPC micelles, DMPC/DHPC and DMPC/CHAPS $q = 0.4$ bicelles. To achieve the most native conditions, we took only the phosphatidylcholine-based lipids and detergents, the proteins were studied at neutral pH and 50 mM ionic strength of the solution. To mimic the charge of the cell membrane, we added LPPG to the LPPC micelles and DMPG to the bicelle samples (Supplementary Fig. 1). Since bicelles were shown to contain a patch of the bilayer[31,32], they were selected for further studies, provided that the spectrum of reasonable quality was obtained. Thus, TLR2tmjm, TLR5tmjm, and TLR9tmjm were first studied in DMPC/DHPC environment. In turn, DPC is known to cause structure distortion in helical membrane proteins[33], therefore it was considered only if all other mimetics fail to provide the necessary spectrum quality and sample stability. In the case of TLR3tmjm, only the DPC micelles provided the NMR spectra of sufficient quality.

### Juxtamembrane regions of TLR2, TLR3, TLR5, and TLR9 form a surface-associated α-helix

The spatial structures of all four objects were studied using the uniform approach, which involved the chemical shift assignment

(Supplementary Figs. 2–5), analysis of chemical shifts in TALOS-N software to get the secondary structure, then analysis of the nuclear Overhauser effect (NOESY) spectra to get the interproton distances and, finally, measurement of residual dipolar couplings (RDCs) in the anisotropic environment, formed by the G-tetrad DNA liquid crystals[34]. Besides, the cross-correlated relaxation of amide groups was utilized to assess the intramolecular mobility of the proteins[35]. Initial analysis revealed that the lifetime of TLR9tmjm in bicelles does not exceed 3–4 days, which is insufficient for the structure elucidation, and the spatial structure of the protein was resolved in LPPC/LPPG micelles. Similarly, the quality of NMR spectra of TLR5tmjm in bicelles was low – signals of residues 654–662 that include parts of TMD and JM regions were not observed, most likely due to the conformational exchange. The spatial structure of this protein was resolved in DPC micelles since LPPC/LPPG did not provide the NMR data of reasonable quality. Thus, as a final result, we obtained the spatial structure of TLR2tmjm in DMPC/DMPG/DHPC 3.2:0.8:10 bicelles, of TLR3tmjm and TLR5tmjm in DPC micelles, and of TLR9tmjm in LPPC/LPPG 3:1 micelles. The NMR data are shown in Fig. 2 (TLR2tmjm) and in Supplementary Figs. 6–9, and are summarized in Table 1.

The resulting spatial structures were quite similar but revealed several important discrepancies (Fig. 3). The structure of TLR2tmjm includes two distinct helical regions (M584-R611 and L615-A627) that are connected by a loop (F612-G614). The N-terminal helix reveals a bend and its length allows assigning it to the TM domain. The second helix is oriented at 80° ± 10° with respect to the transmembrane one and is most likely associated with the bicelle surface. The structure of TLR9tmjm is almost identical to the one of TLR2. Two helices are formed in the regions W816-L838 and L843-A852, the angle between the helices equals 60° ± 20° (Supplementary Fig. 10). NMR spectra of TLR9tmjm were analyzed in all three membrane mimetics (Supplementary Fig. 11). Secondary chemical shifts of transmembrane

**Table 1 | Statistics of NMR input data and parameters of the obtained spatial structures**

| NMR distance, dihedral, and RDC constraints | TLR2tmjm | TLR3tmjm | TLR5tmjm | TLR9tmjm |
|---|---|---|---|---|
| Distance constraints | | | | |
| Total NOE | 363 | 215 | 471 | 344 |
| Intra-residue | 256 | 114 | 323 | 130 |
| Inter-residue | 107 | 101 | 148 | 214 |
| Sequential ($|i-j| = 1$) | 61 | 79 | 105 | 103 |
| Medium-range ($|i-j| \leq 4$) | 46 | 22 | 43 | 111 |
| Long-range ($|i-j| > 5$) | 0 | 0 | 0 | 0 |
| Hydrogen bonds | 0 | 0 | 0 | 15 |
| Total dihedral angle restraints | | | | |
| φ | 44 | 33 | 47 | 36 |
| ψ | 44 | 34 | 47 | 38 |
| $\chi_1$ | 12 | 10 | 11 | 9 |
| RDC restraints | | | | |
| $D_{NH}$ | 28 | 40 | 17 | 41 |
| $D_{NH}$ range (Hz) | −39.0..5.2 | −17.3..6.7 | −39.5..28.8 | −6.5..10.0 |
| RQC of $D_2O$ (Hz) | 15.7 | 12.7 | 14.8 | 36.5 |
| Constraints per residue | | | | |
| **Structure statistics (for the set of 10 best structures)** | **TLR2tmjm** | **TLR3tmjm** | **TLR5tmjm** | **TLR9tmjm** |
| CYANA target function | 0.39 ± 0.06 | 1.42 ± 0.10 | 1.16 ± 0.17 | 1.27 ± 0.04 |
| Violations | | | | |
| Distance constraints (Å) | 0.24 ± 0.00 | 0.25 ± 0.02 | 0.28 ± 0.08 | 0.40 ± 0.02 |
| Max. distance constraint violation (Å) | 0.24 | 0.28 | 0.48 | 0.45 |
| Dihedral angle constraints (°) | 1.0 ± 0.3 | 0.98 ± 0.78 | 1.27 ± 0.66 | 2.97 ± 0.52 |
| Max. dihedral angle violation (°) | 1.51 | 3.26 | 2.28 | 4.17 |
| RDC constraints (Hz) | 0.19 ± 0.10 | 2.48 ± 0.03 | 0.21 ± 0.08 | 0.63 ± 0.33 |
| Max. RDC constraint (Hz) | 0.34 | 2.53 | 0.34 | 1.47 |
| Average pairwise r.m.s. deviation for secondary structures elements (α-helix regions) | H586-G606(TM) L615-Q626 | F702-M707 T710-G727(TM) S737-L742 | V634-F643 T647-R664(TM) C667-L677 | C818-M834(TM) L843-W853 |
| Backbone (Å) | 0.45 ± 0.19 0.54 ± 0.24 | 0.47 ± 0.11 0.29 ± 0.22 0.52 ± 0.23 | 1.12 ± 0.53 2.15 ± 0.43 0.78 ± 0.22 | 0.50 ± 0.19 0.33 ± 0.07 |
| Heavy (Å) | 1.27 ± 0.11 1.39 ± 0.25 | 2.15 ± 0.38 0.93 ± 0.13 1.47 ± 0.42 | 1.93 ± 0.53 2.83 ± 0.35 1.64 ± 0.35 | 1.04 ± 0.28 1.59 ± 0.38 |
| *PDB ID* | 8AR0 | 8AR1 | 8AR2 | 8AR3 |

and juxtamembrane helices are almost identical, indicating that the same structure is adopted by TLR9tmjm in DMPC/DHPC bicelles, with the differences observed only at the interhelical hinge region.

Unlike other proteins under investigation, TLR3tmjm forms three helical segments. The TM helix encompasses residues F701-W734 and is kinked strongly at N709 and bent at G727. The JM helix is formed in the region S737-L742 and is oriented at 65° ± 30° with respect to the TM domain. Finally, the structure of TLR5tmjm is the most peculiar. According to the chemical shift data, TLR5tmjm forms an extra long 48-residue α-helix, which encompasses almost the whole protein, and contains no obvious kinks (Supplementary Fig. 7). RDC data reveals that the helix is bent, forming a U-shaped structure, with clearly observed transmembrane and juxtamembrane parts. Analysis of secondary chemical shifts, observed in bicelles and LPPC/LPPG micelles (Supplementary Fig. 11) suggests that this conformation is not an artifact caused by the use of DPC micelles. The same secondary structure is found in other membrane mimetics (probably, except for the bend of the helix at the N-terminus, which is not preserved in bicelles), and the slow motions of the bent helix explain the poor quality of TLR5tmjm NMR spectra.

## Juxtamembrane regions of TLRs interact with lipids and are immersed into the bilayer

As a next step, we utilized NMR spectroscopy to analyze the protein-lipid interactions in the object under investigation. This study was performed for TLR2tmjm and TLR9tmjm because these two proteins were explored in the presence of either lysolipids or bilayer lipids, which were taken in the non-deuterated form. We utilized the nuclear Overhauser effect spectroscopy to directly measure the inter-molecular contacts between the protein backbone amide groups and lipid $CH_2$ groups (at -1.3–1.25 ppm) and additionally quantified the water exchange cross-peaks. Such an approach provides a contrast between the lipid-exposed and water-exposed regions. According to our data, both proteins behave similarly and juxtamembrane domains reveal contacts with the lipid acyl chains, with intensities, comparable to the TM domains. Interactions with water are observed mainly at the N- and C-terminal and in the interhelical hinge region (Fig. 4). This suggests that JM domains are not just surface associated but are immersed deeply in the hydrophobic part of the bilayer, while the interhelical hinge region is water-exposed.

Since TLR2tmjm is stable in phospholipid bicelles, we investigated the effect of the lipid environment on the structure of its juxtamembrane regions. Variation of the bicelle size ($q = 0.4$–$0.7$) did not

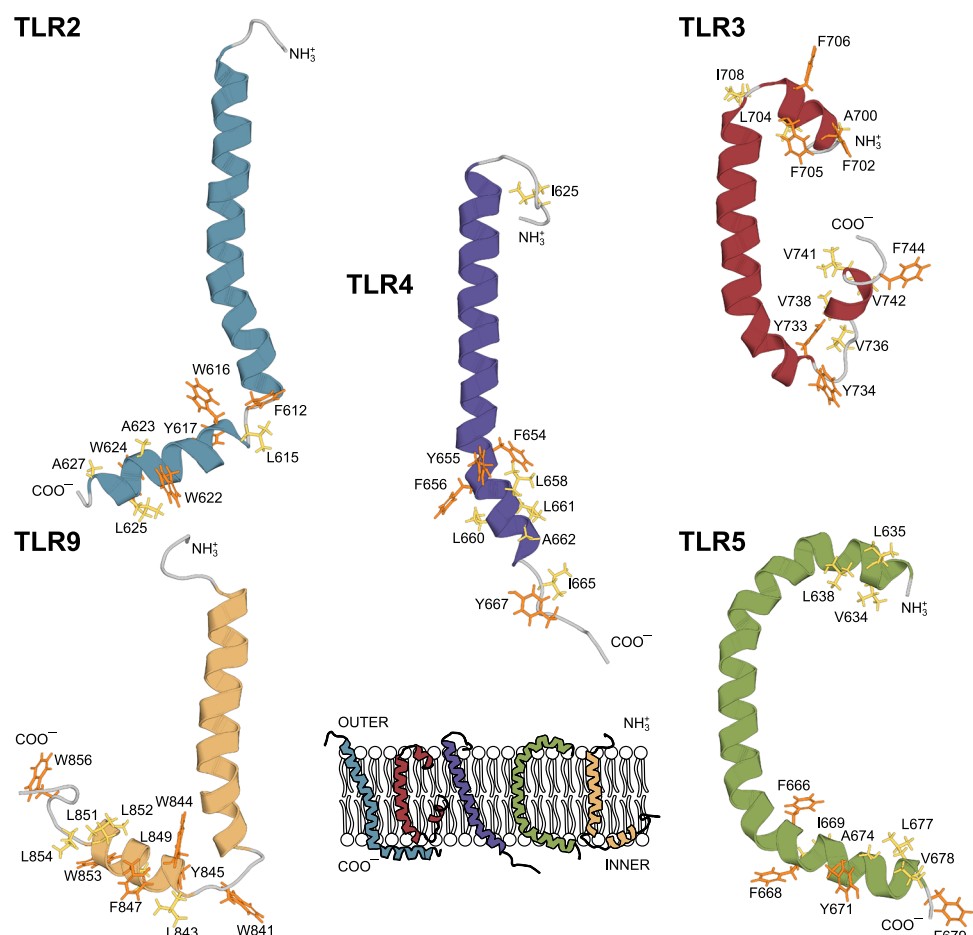

**Fig. 3 | Spatial structures of TLR transmembrane domains with adjacent juxtamembrane regions, as obtained in the current work.** The structure of TLR4 is taken according to the work by Mineev et al.[30] Bulky hydrophobic and aromatic side chains in the juxtamembrane parts of TLRs are shown in yellow and orange, respectively.

substantially affect the NMR spectra of the protein, indicating that the bicelle size is sufficient for the correct positioning of the JM domain (Supplementary Fig. 12). The addition of up to 20% of anionic lipids to the bicelles also did not alter the structure of juxtamembrane regions (Supplementary Fig. 13). Finally, as the cell membrane is inhomogeneous and TLR2 can migrate between the various membrane compartments[36], we studied the effect of the bicelle thickness on the TLR2tmjm (Supplementary Fig. 14). Using our previously established approach[37,38], we showed that DPPC/DHPC mixtures follow the ideal bicelle model (Supplementary Fig. 15) and measured the NMR spectra of TLR2tmjm in this environment. We observed that the spectra in thick and thin bicelles are almost identical and the difference is observed mainly in the hinge region and the adjacent turns of the TM and JM helices (Supplementary Fig. 14). Thus, the protein adapts to the bilayers with different thicknesses by variating the mutual arrangement of TM and JM helices, while the structure of the two domains is left unchanged.

## The structure of TLR transmembrane and juxtamembrane domains is retained in lipid bilayers as revealed by in silico experiments

As the structures were obtained in membrane mimetics, it was important to analyze their behavior in membranes. This was done using 1000 ns MD simulations in explicit membranes. Simulations have shown that in all cases TLRtmjms generally retain their secondary structure (Fig. 5a). During MD, two processes were observed. First, adapting to the nonpolar environment, the TM helices changed their tilt to the membrane. The angle to the membrane normal depended on the length of the TM helix − for TLR2 it was 30 degrees, and for other molecules, it was about 60 degrees. During MD, the short N-terminal helices detected in TLR3 joined to the TM helix after the first 50 ns of dynamics; however, this long TM helix was not stable, as revealed by the final part of the trajectory. In the case of TLR5, the N-terminal bend in the TM helix was observed till the middle of the trajectory and then disappeared, in agreement with the NMR chemical shifts of TLR5 in bicelles (Supplementary Fig. 11). Second, JM helices rotated along their axis to find the optimal location of charged and nonpolar residues at the lipid-water interface. As in the experiment, after adaptation, all fragments of the studied peptides were in contact with the lipid tails. To analyze the behavior of peptides in different environments, we compared the internal mobility maps (Fig. 5b, c). These maps display the stability of distances between the CA atoms of protein residues, reveal the protein parts that move synchronously or independently during the simulations and thus illustrate the structural domains in peptides. The spatial structures obtained by NMR reveal several domains for each peptide (Fig. 5b). These domains mostly correspond to α-helices (blue squares on the maps). In the NMR sets, these helices have different relative positions or orientations (red areas) adapted to interact with micelles or bicelles. In the MD simulation, the structural behavior was similar, as illustrated by the mobility maps (Fig. 5c). Thus, one can conclude that the peptides adapt to the membrane environment utilizing the structural domains found in the experiment. It validates the used experimental systems and points out that structure evolution in the different lipid-like environments goes in similar ways.

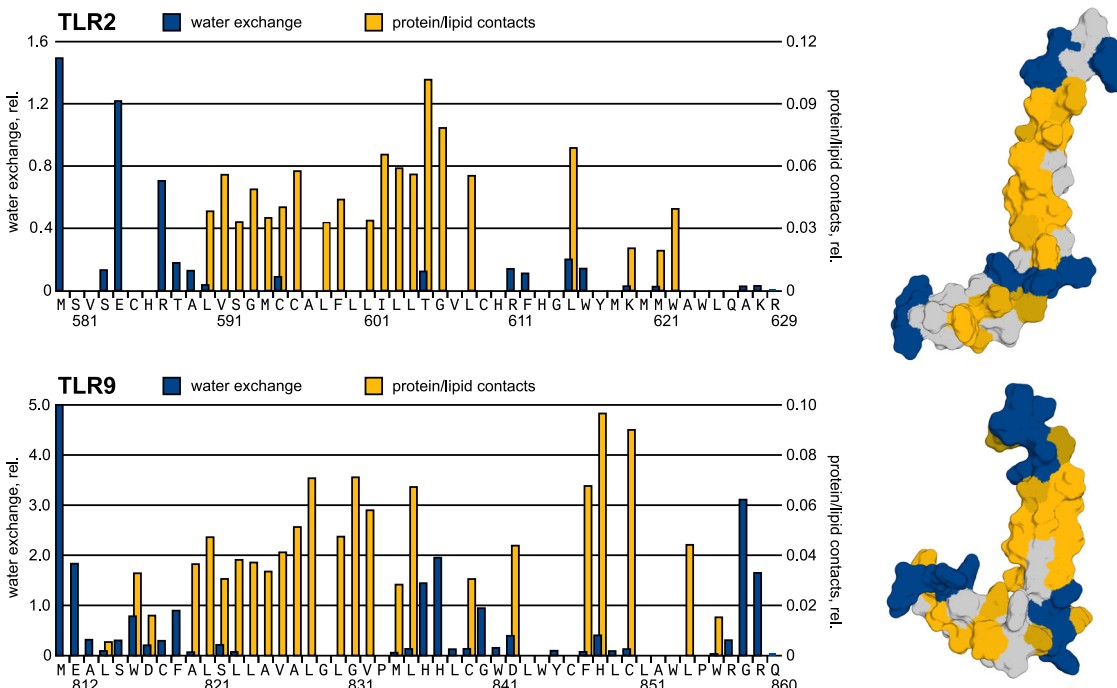

**Fig. 4 | Interaction of TLR2tmjm and TLR9tmjm with lipid and aqueous environment.** Bar plots indicate the intensities of NOE peaks between the amide protons and CH$_2$ protons of lipids (yellow) and NOE exchange peaks between the amide protons and water (blue). Each value was normalized to the intensities of diagonal peaks. On the right, the surfaces of TLR2tmjm and TLR9tmjm are shown. Residues, interacting with water and with lipids are painted blue and yellow, respectively. The gray color indicates that there is no information available. Source data are provided as a Source Data file.

The structures obtained in the result of simulations are shown in Fig. 5d.

### A correct sequence of TLR2 juxtamembrane regions is necessary for receptor activation

Since all the TLRs were found to contain the juxtamembrane helices, we hypothesized that these regions could be involved in the activation mechanism. To test this assumption, we investigated the role of the JM region in the activation of TLR1/TLR2 receptor in HEK Blue 293 cells, monitoring the activity of NF-κB by Phospha-Light secreted embryonic alkaline phosphatase (SEAP) reporter gene assay system after the stimulation of the receptor with its specific ligand, Pam3CSK4. To study the role of the JM region, we designed several chimeric proteins: (1) TLR2 with deleted JM region (TLR2dJM); (2) TLR2 with the JM region substituted to the flexible (GGS)$_n$ linker of the same length (TLR2GGS); (3) TLR2 with the sequence of JM region being randomly scrambled (TLR2scr); (4) TLR2 with the JM helix transplanted from TLR1 (TLR2JM1); and four similar constructs of TLR1: TLR1dJM, TLR1GGS, TLR1scr, and TLR1JM2 (Fig. 6). Expression and transmembrane localization of all the designed constructs were not different from the wild-type receptor as revealed by the real-time PCR and flow cytometry (Supplementary Figs. 16, 17). According to the functional assay, all the modifications that were introduced to the JM of the TLR2 protein abolished the ligand-induced activation of the TLR1/TLR2 receptor (Fig. 6). Therefore, not just the presence of a hydrophobic helix with a certain length, but also the correct order of amino acid residues in the JM helix of TLR2 is necessary for the proper activation of TLR1/2. On the other hand, all the variants of TLR1 were as active as the wild-type protein, so the JM region of TLR1 is not that important.

## Discussion

To summarize, here we solved the structures of the transmembrane and juxtamembrane hydrophobic regions of four TLRs − TLR2, TLR3, TLR5, and TLR9. To our regret, we did not manage to determine all the

structures in lipid bicelles, which are known to provide the most native-like environment, and structures of two proteins were obtained in alkyl-phosphocholine micelles (DPC) that may cause distortions in the structures of helical membrane proteins[33]. On the other hand, we would like to point out that three out of four proteins (TLR2, TLR5, and TLR9) were actually investigated in DMPC/DMPG/DHPC 3.2:0.8:10 bicelles, despite the fact that the structures of TLR5 and TLR9 were not resolved due to the sample stability and NMR spectrum quality problems. For TLR5 and TLR9 we obtained complete or almost complete NMR chemical shift assignments in all three membrane mimetics that were used in the work: DPC, LPPC/LPPG, and bicelles, which allowed us to analyze the effect of the environment on the protein structure (Supplementary Fig. 11). Besides, all the obtained structures were adapted to the lipid bilayer in full-atomic MD simulations. As we show, TLR9 provides almost identical chemical shifts in all three tested mimetics and is stable in MD simulations, which suggests that the obtained structure is likely to be retained in bilayers. In the case of TLR5, we observe the mimetic-dependent chemical shift differences at the N-terminus, which indicate that the N-terminal bend of the helix may not be retained in bicelles and, therefore, in bilayers. However, the chemical shifts of the C-terminal helix are also identical in all three mimetics, thus suggesting that the cytoplasmic helix is the element, which does not depend on the type of membrane-like environment. In the case of TLR3, we have only the MD simulations to judge the relevance of the obtained structure. The N-terminal kink of the transmembrane helix is not retained in MD simulations, but the conformation of the cytoplasmic juxtamembrane region is preserved. Therefore, we chose to focus not on the differences between the four structures that could be caused by the membrane mimetic but to discuss the similarities that persist in various membrane mimetics and in MD simulations.

Thus, we can state that despite the certain discrepancies, all four proteins under investigation share an important common feature as revealed by the experiments. In all the TLRs we observe the presence of

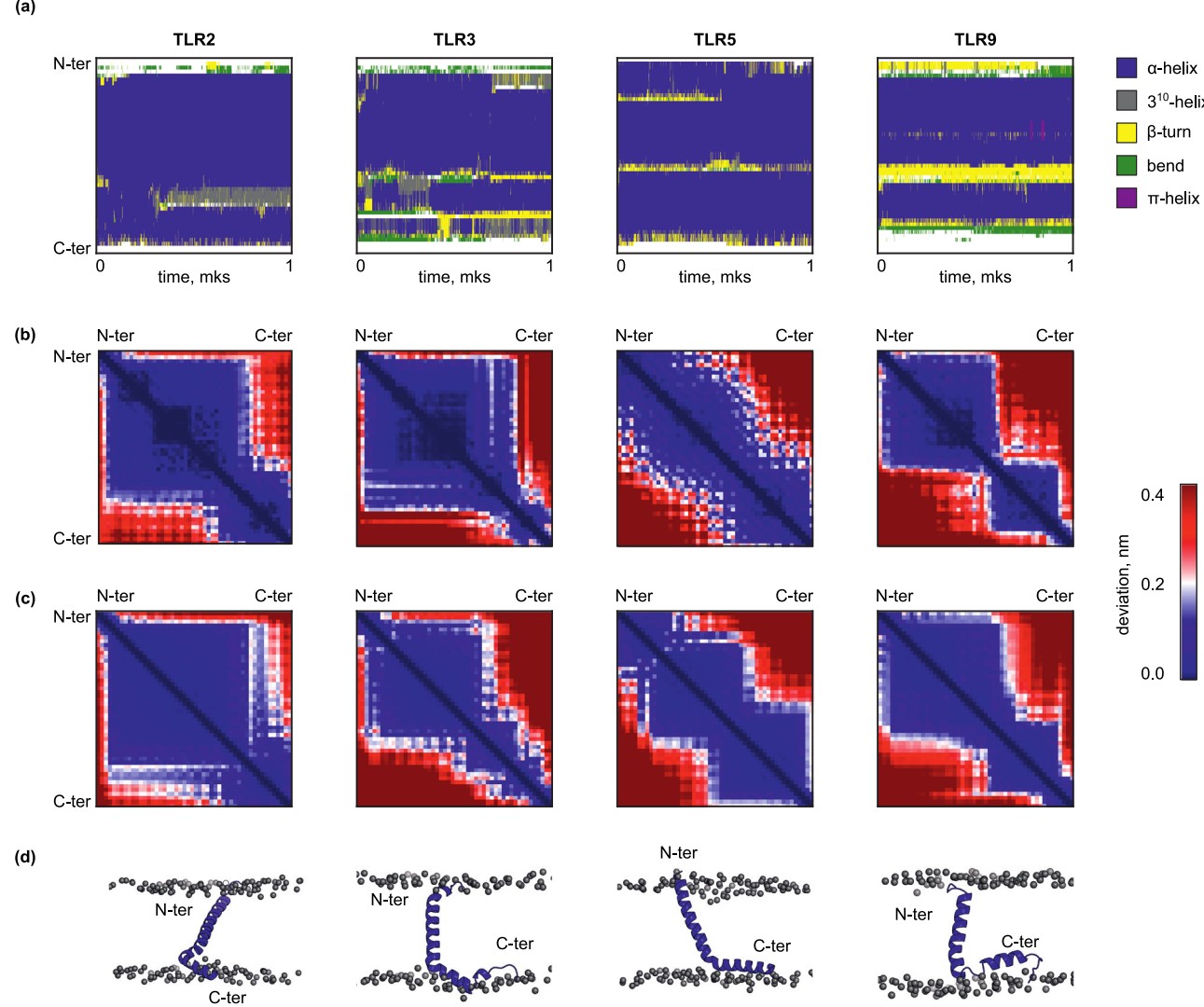

**Fig. 5 | Adaptation of TLRtmjm to membrane environment. a** Evolution of the secondary structure in MD. The colors of the elements of the secondary structure are given on the right. **b**, **c** Internal mobility maps, calculated by the experimental (**b**) and the simulation (**c**) sets of structures. Deviation (nm) indicates the standard deviation of the distance between the CA atoms of corresponding residues, calculated for the set of NMR structures or frames of MD trajectory. The color scale is shown on the right. **d** Structures of studied proteins after MD relaxation. The locations of lipid P atoms are shown by gray spheres. Proteins are given in ribbon representation.

a cytoplasmic juxtamembrane α-helix. The length of the JM helix varies from three turns in TLR2 and TLR5 to only 6-8 residues in TLR3. Nonetheless, it is now obvious that the linker that connects the globular TIR domain to the membrane is quite short and includes 12-14 residues. In all the structures, the juxtamembrane helix is membrane-associated, and for TLR2 and TLR9 it was directly shown to be immersed into the membrane interior, as revealed by the experimentally observed protein-lipid contacts (Fig. 4). These cytoplasmic helices are not amphipathic, and are almost entirely hydrophobic: there is no pronounced asymmetry between the polar and hydrophobic residues, and the contents of polar residues are relatively low. However, the analysis suggested by White and Wimley[39] reveals that intracellular juxtamembrane domains of TLRs are rich in amino acids that favor the lipid/water interfaces (e.g. Trp, Phe, and Tyr residues), which explains their position at the membrane surface (Supplementary Fig. 18).

The obtained results need to be discussed in regard to the other experimental data, reporting on the structure of TLR membrane domains. These data include the NMR structure of TLR3 TM domain[29] and cryo-EM structures of TLR3 and TLR7 in complex with a chaperone[28]. Both cryo-EM structures lack the juxtamembrane and

cytoplasmic domains, either due to high mobility or the disordered state. On the other hand, the end of the TM domain in both cases corresponds to the TMD length in our structures of TLR3tmjm and TLR9tmjm, which supports the observed position of the JM helix. In our hands, the position of the JM domain with respect to the TMD is not fixed and that could explain the absence of the JM region in the cryo-EM density maps.

Both the NMR and cryo-EM structures of TLR3 TMD do not reveal the N-terminal kink in the transmembrane helix that was observed in the current study. However, this part of the protein was found to be destabilized in the monomeric state of TLR3 TMD and in detergents with the low length of the acyl tail[29]. This kink occurs at Asn709, most likely the presence of polar Asn residue inside the membrane is not favorable. We observed that the kink was abolished during the MD simulations, and it is not well compatible with the possible presence of the extracellular domain of TLR3. Therefore, most likely the N-terminal kink in the TM helix of TLR3 is an artifact of the chosen membrane mimetic. The absence of the kink in bicelles and thick micelles could explain the high oligomerization propensity of TLR3tmjm in these membrane mimetics since Asn709 was reported to promote the

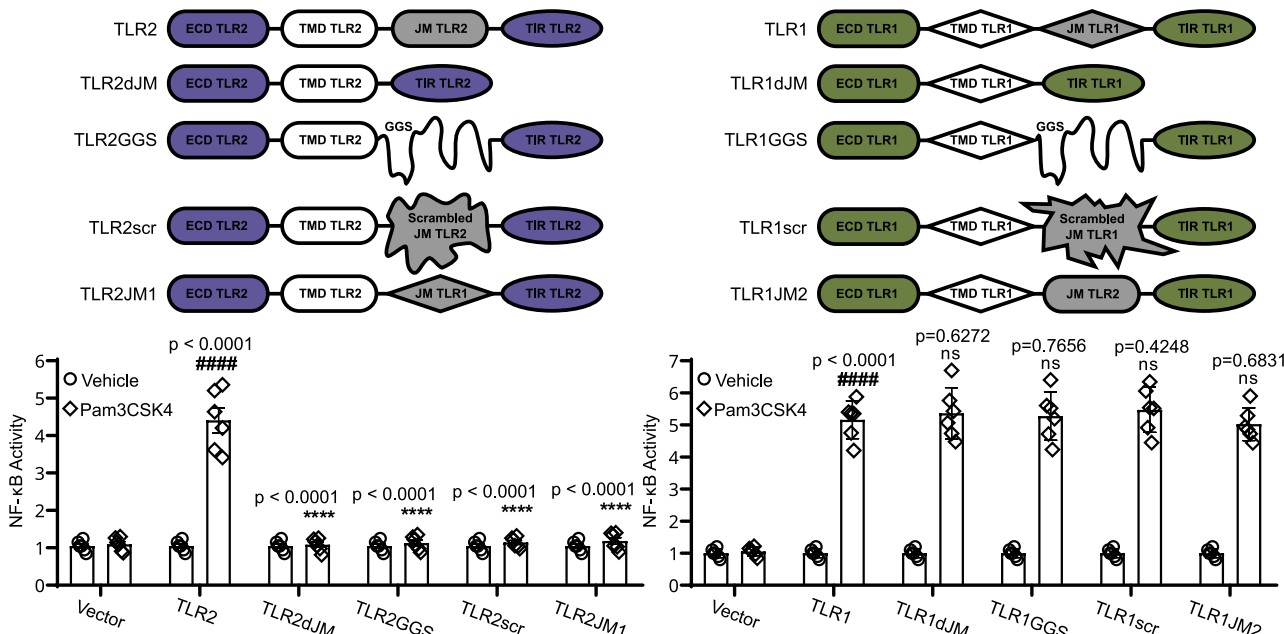

**Fig. 6 | Functional role of JM helix in the functioning of TLR1/TLR2 receptor.**
Shown is the NF-κB activity measured for TLR1 and TLR2 constructs upon stimulation with Pam3CSK4 ($n = 6$). Data are represented as mean ± SD. Statistical significance according to the unpaired two-tailed Student's $t$ test is indicated as

follows: ****$p < 0.0001$ with respect to the positive control, ####$p < 0.0001$ with respect to the negative control. ns denotes that changes with respect to the positive control are not significant. Source data are provided as a Source Data file.

high-order oligomerization of TLR3 TMD[29]. Structures of other TMDs and the juxtamembrane part of TLR3 are all stable during MD, thus the presence of the cytoplasmic JM helices does not raise doubts and should be considered as a common feature for all the TLRs except TLR4.

Analysis of TLR evolution reveals that the observed juxtamembrane helices are conservative and the average rate of mutations is comparable with TIR domains (Supplementary Fig. 19). It suggests that the juxtamembrane domain plays an important role in TLR activation, and is in good agreement with the results of our functional assays. According to the present work, the substitution of a JM helix of TLR2 with a flexible linker blocks the activation of TLR1/TLR2 receptor, therefore it is unlikely that the function of a JM domain is a correct spacing between the membrane and TIR domain of TLRs. Moreover, random scrambling of amino acids in the JM helix of TLR2 or substitution of the helix with the one of TLR1 also inhibit the full-length receptor. This implies that the correct positioning of TIR domains with respect to the membrane is also not the major role of the region, the proper arrangement of certain amino acids is as well very important. In contrast, we observe that all the manipulations with the JM helix of TLR1 do not affect the activation of the TLR1/TLR2 receptor. Such an asymmetry in the roles of JM regions played in the activation of TLR1/TLR2 may be explained by the different functions of intracellular domains. According to the coimmunoprecipitation studies, only the TIR domain of TLR2 is capable of binding the adapter protein MyD88[40], therefore one could assume that the JM helix is important for the recruitment of adapter proteins but not for the dimerization of the receptor TIR domains. Similar results can be found in literature describing the essential role of the JM helix in TLR3[41]. Residues F732, L742, and G743 of the receptor (JM region) were found important for the ligand-induced NF-κB and IFN-β promoter activation by site-directed mutagenesis.

Based on the obtained data, we could propose several possible functions of TLR juxtamembrane regions. (1) JM regions could encode the binding sites for specific lipids and can be used to deliver the TLRs to certain compartments of the cell membrane; (2) JM regions could

promote the specific interactions with the adapter proteins of TLRs, e.g. with the membrane-binding domain of TIRAP[42]; (3) JM regions may interact directly with the TIR domains and TLRs, regulating their activation. Therefore, we could hypothesize that JM helices are expected to change their conformation upon the ligand-induced dimerization of TLRs, regulating the state of their globular TIR domains. These changes can be coupled either to the direct interactions between the TIR and JM domains, interaction with the adapter proteins, membrane adsorption/desorption of TIR domains, or to the trafficking of TLRs to certain compartments of the cell membrane, e.g. lipid rafts[36,43,44]. Elucidation of the role played by the JM regions in TLR activation should be a subject of further investigation.

Together with the previously published structure of TLR4[30], we can state that for all five TLR subfamilies, conformations of the membrane-associated parts of receptors are now available, however, in a membrane mimetic environment and within the truncated protein constructs. This apparently closes one of the major "blank spots" in TLR structural biology, structures of all three TLR domains are now provided by the "divide and conquer" approach for several proteins. TLR2 receptor becomes the most thoroughly investigated—it is the only TLR with the structures of all three domains being experimentally resolved.

## Methods
All reagents were provided by Sigma Aldrich, unless otherwise specified.

### Cell-free synthesis of TLR fragments
The genes encoding transmembrane and juxtamembrane residues of TLRs were amplified by PCR (residues 581-629 UNIPROT ID O60603, TLR2tmjm; residues 698-746 UNIPROT ID O15455, TLR3tmjm; residues 632-680 UNIPROT ID O60602, TLR5tmjm; residues 812-860 UNIPROT ID Q9NR96, TLR9tmjm). The PCR products were cloned into the pGEMEX1 vector using ligation by NdeI and HindIII restriction sites. The final construct was confirmed by DNA sequencing. All protein fragments were expressed as precipitate using the cell-free production

system based on the *E.coli* lysate[30]. A standard cell-free reaction (3 ml of RM) was carried out in 50 ml tubes using Pur-A-Lyzer Maxi dialysis kit (#PURX35050). The FM:RM ratio was 12:1 and mixture contained 100 mM HEPES (#H4034) at pH 8.0, 0.83 mM EDTA (#E9884), 0.1 mg/ml folinic acid (#47612), 20 mM acetyl phosphate (#A0262), 1.2 mM ATP (#1191, Calbiochem) and 0.8 mM each of GTP (#51120)/CTP (#30320)/UTP (#94370), 2 mM 1,4-dithiothreitol (#D9779), 0.05% sodium azide (#31803515, Molekula), 2% PEG-8000 (#89510), 20 mM magnesium acetate (#M5661), 270 mM potassium acetate (#1131, Gerbu), 60 mM creatine phosphate (#27920), 1 mM each of 20 amino acid, 0.2 tablet of complete protease inhibitor (#43203100, Roche). The RM mixture additionally contained 0.5 mg/ml *E. coli* tRNA (#12699020, Roche), 0.25 mg/ml creatine kinase from rabbit muscle (#10127566001, Roche), 0.05 mg/ml T7 RNA polymerase, 0.1 U/μl Ribolock (#E00384, Thermo Scientific), 0.02 μg/μl plasmid DNA, and 30% S30 CF extract. Reactions were conducted overnight at 32 °C and 150 rpm in an Innova 44 R shaker (New Brunswick). Amino acid mixture of 15 N (#NLM-6695, CIL) or 13 C/15N-labeled (#CNLM-6696, CIL or #CCN070P1, Cortecnet) amino acids were used to obtain a uniformly 15 N or 13 C/15N-labeled protein sample, respectively.

The TLR9tmjm and TLR2tmjm proteins were purified using size-exclusion chromatography. The precipitate from the cell-free reaction was washed 2 times with aqueous buffer (20 mM Tris 8.0, 250 mM NaCl) and solubilized with 500 μl of buffer (20 mM Tris (#1.08387, Merck), pH 8.0, 250 mM NaCl (#1112, Gerbu), 2% lauryl sarcosine (#8.14715 Merck), 15 mM bME (#M6250)). After centrifugation (for 60 min at 25,000 × *g* at room temperature) the clarified protein solution was applied onto a 10/600 Tricorn column packed with Superdex 200 prep grade (GE Healthcare) resin and pre-equilibrated with SEC-buffer (20 mM Tris, pH 8.0, 50 mM NaCl, 0.5 % lauryl sarcosine, 10 mM bME). Protein-containing fractions were combined and precipitated by a TCA/acetone procedure[45]. TCA (#1.00807, Merck) was added to protein solution up to 10%, the sample was frozen at −20 °C (15–20 min) and the protein was precipitated by centrifugation for 15 min at 13,000 × *g* at 4 °C. The precipitate was washed 3 times with acetone (#Acetone, Chimmed) to remove the detergent.

The precipitate from the cell-free reaction for TLR3tmjm and TLR5tmjm proteins was washed 2 times with an aqueous buffer (20 mM Tris 8.0, 250 mM NaCl) and used directly for sample preparation (without additional purification).

## Sample preparation
Dry samples of proteins were dissolved in TFE (Trifluoroethanol) (#D027BB, Eurisotop)/water 2:1 v/v mixture, with the addition of 3–5 mM TCEP (Tris(2-carboxyethyl)phosphine) (#C4706). The obtained solutions were incubated in a shaker and an ultrasonic bath for 15-25 min at room temperature and centrifuged at 8300 *g* for 7 min in an Eppendorf MiniSpin. Then the supernatants were supplied with the aqueous stock solutions of lipids and detergents and water up to the TFE:H₂O volume ratio 1:1 and freeze-dried. The resulting powder was dissolved in the NMR buffer (20 mM NaPi, 0.05% NaN₃, 5% D₂O (#SD-19, SIC)), and incubated in a shaker and ultrasonic bath for 15–25 min at room temperature. The protein solution was centrifuged at 8300 *g* for 5–7 min in an Eppendorf MiniSpin and a supernatant was used for NMR experiments. pH was controlled and adjusted at the final step.

Samples for the RDC measurements were prepared as follows: weighed powder of dGpG was dissolved in the NMR-RDC buffer (20 mM KiP, 0.05% NaN3, 5%D2O, 100 mM KCl (#P3911), pH=7.0). The amount of dGpG was chosen so that the concentration of dGpG in the final sample was 25–28 mg/ml. The formation of the liquid crystal medium was checked by measuring the RQC of D₂O. Then the sample was freeze-dried. There is an option to make dialysis before freeze-drying to decrease the concentration of salts as mentioned in ref. [46]. Then the resulting powder was dissolved in 95%H₂O/5%D₂O with an

appropriate volume and added to dried powder of a 15N-labeled protein with lipids and detergents that was prepared as described above. The resulting solution was incubated for several hours in the shaker at 40–45 °C. pH was controlled and adjusted at the final step. If the formation of liquid crystal did not occur, we added the extra amount of dGpG.

## Synthesis of dG-p-dG
The precursor of dinucleotide dG-p-dG − fully protected dinucleotide 5′-O-(DMTr)-N²-iBuG-3′-O-P(O)(OClPh)N²-iBuG-3′-O-Ac − was synthesized in accordance with[47] and isolated by chromatography on silica gel. After full deprotection of this compound, the target dinucleotide was isolated by anion-exchange chromatography and purified by reversed-phase chromatography.

## NMR spectroscopy
NMR experiments were performed on the Avance III 600, 700, and 800 MHz spectrometers (Bruker Biospin, Germany) at 45 °C. ¹H, ¹³C, and ¹⁵N assignments were obtained in CARA 1.9.3 via the standard procedure based on 3D HNCO, 3D HNCA, 3D HNCOCA, and 3D ¹H¹⁵N-NOESY-HSQC spectra, triple resonance spectra were recorded using the BEST-TROSY approach[48]. Signals of protein methyl groups were assigned using the constant-time versions of HCCH-TOCSY with FLOPSY mixing[49]. Aromatic side chains were assigned utilizing the TROSY-hCCH-COSY[50] and (Hb)Cb(CgCC)H experiment[51]. A non-uniform sampling of indirect dimensions was applied when necessary, such spectra were processed with qMDD 3.2 software[52].

Protein spatial structures were calculated in CYANA 3.97.8 software, starting from 100 random initial positions[53]. The backbone torsion angle restraints were obtained in the TALOS-N 4.12 software, based on the NMR chemical shifts[54]. Sidechain torsion angle restraints were obtained by the manual analysis of NOESY spectra and ³J_{C_γC′}, ³J_{C_γN} couplings measured in the spin-echo difference ¹H¹³C-HSQC experiments[55,56]. Upper distance restraints were obtained from the intensities of cross-peaks 3D ¹H¹⁵N-NOESY-HSQC, 3D ¹H¹³C-NOESY-HSQC, and 3D ¹H¹³C-NOESY-ct-HSQC. Mutual orientations of NH-bonds were obtained from the ¹D_{NH} RDCs, measured using the IPAP-¹H¹⁵N-HSQC[57]. RDC restraints were applied at the final stage of the structure calculation. The structure was refined iteratively three-four times with the fitting of alignment tensor parameters. Spatial structures were analyzed in MOLMOL 2K2.0[58] and PyMOL 2.5.2 (Schrödinger) software. Conditions that were chosen for the structure determination of all four proteins are presented in Supplementary Table 1.

To assess the internal mobility of the peptides, the cross-correlated ¹H/¹⁵N relaxation rates were measured and converted into the correlation time of rotational diffusion as described[35]. To measure the protein/lipid interactions, the intensities of the NOE cross-peaks at 1.35–1.25 ppm (lipid CH₂ groups) were quantified and normalized to the intensities of corresponding diagonal signals. Residues with their own protons or protons of i-1 amino acid resonating at the close chemical shifts were excluded from the analysis. To assess the water exchange rates of the protein amide groups, the intensities of signals at 4.6 ppm (H₂O) were normalized to the intensities of corresponding diagonal signals. To quantify the effect of bilayer thickness and of the charged lipids on the structure of proteins, we utilized chemical shift perturbations (CSPs). CSPs were calculated using the equation:

$$CSP = \sqrt{\Delta CS(H\,ppm)^2 + \Delta CS(N\,ppm)^2/10^2} \qquad (1)$$

where $\Delta CS$ denotes the corresponding chemical shift changes, and ¹⁵N chemical shifts are scaled by the factor of 10, to take into account the different gyromagnetic ratios of ¹H and ¹⁵N nuclei[59].

To assess the bicelle formation in the DPPC/DHPC mixture, we used our previously published approach that relies on the lipid

diffusion measurement by NMR[37,38]. The initial q = 0.9 sample was titrated by DHPC, and diffusion coefficients of lipid and detergents were measured using the DSTE-WATERGATE experiment[60] with convection compensation at 45 °C, pH 7.0.

## MD simulations

The properties of TLR2, TLR3, TLR5, and TLR9 in the membrane were analyzed using molecular dynamics.

All simulations were conducted using GROMACS 2021.5[61]. We used amber14SB-based topology for proteins[62], Slipids molecular topology for lipids[63], and the tip3p water model[64]. Electrostatic interactions were treated using the particle-mesh Ewald summation with fourth-order spline interpolation[65]. The initial cutoff value of 1.2 nm and Ewald grid spacing of 0.12 nm were tuned during the CPU-GPU loading balance calculations. The MD simulations were conducted in the isothermal-isobaric (NPT) ensemble with an isotropic pressure of 1 bar and a constant temperature of 310 K. The temperature and pressure were controlled using a nose-hoover thermostat[66] and Parrinello–Rahman barostat[67] with 0.5 and 1.0 ps relaxation parameters, respectively, and compressibility of $4.5 \times 10^{-5}\,bar^{-1}$ for the barostat.

The starting configuration of the system was constructed as follows. The bilayer was composed of two monolayers of POPC molecules joined by their tail parts such that the distance between the phosphorus atoms along the membrane normal was equal to 3.4 nm. Each monolayer was composed of 100 molecules, oriented along the membrane normal. The molecules were joined into monolayers such that the atoms from the phosphorus of the lipids were located in the orthoscopic grid knots with 0.8 nm spacing. This bilayer was equilibrated via 10 ns MD. Then, for each protein, we took the best experimental model, aligned its TM helix perpendicular to the membrane, and inserted the peptide into the membrane, removing overlapping lipid molecules. Next, the system was solvated, the water molecules from the bilayer nonpolar part were removed, and $Na^+$ and $Cl^-$ ions were supplemented to render the system electrically neutral and emulate the solvent NaCl concentration equal to 0.1 mM. The resulting systems were equilibrated in energy minimization followed by the 10 ns MD with 1 fs integration step with fixed peptide position. Then, 500 ns production runs were calculated with 2-fs integration steps.

The secondary structure was calculated using Gromacs utility. The internal mobility maps were created from the standard deviation of the distances between CA atoms, calculated for the set of experimental structures or for the set of MD frames with 1 ns timestep. The helix axis was calculated by aligning the CA atoms of the participating residues using the least squares method.

## Secreted embryonic alkaline phosphatase (SEAP) assay

HEK Blue 293 cells were obtained from InvivoGen (hkb-null2). HEK Blue 293 cells were stably transfected with a SEAP reporter gene which was placed under the control of the NF-κB transcriptional response element. The cells were cultured in supplemented DMEM medium (10% FBS, 50 unit/ml penicillin, 50 μg/ml streptomycin, and 1×HEK blue selection). HEK Blue 293 cells were co-transfected with the wild-type or chimeric variant of human TLR1 and TLR2 using Lipofectamine 2000 (Invitrogen) according to the manufacturer's instructions. After 48 h transfection, cells were seeded at a density of $1 \times 10^5$ cells/ml. After 24 h incubation, the medium was replaced with supplemented Opti-MEM medium (0.5% FBS, 50 unit/ml penicillin, 50 μg/ml streptomycin, and 1% of non-essential amino acid (NEAA)). Pam3CSK4 (50 ng/ml) was added and cultured for 8 h. NF-κB activity was measured through the Phospha-Light SEAP Reporter Gene Assay System according to the manufacturer's instructions. All the experiments were repeated six times for the independent samples and analyzed using the directional Student's $t$ test. Gen5 microplate reader and GraphPad Prism 8.0.1 software were used to collect and analyze the data.

## qRT-PCR

HEK Blue 293 cells co-expressing the wild-type or chimera of human TLR1 and TLR2 were seeded at a density of $2 \times 10^5$ cells/ml in 6-well plates. After 24 h incubation, total RNA was extracted, cDNA was synthesized and qPCR was performed[21]. Analytik Jena qPCRsoft 4.0 was used to collect and analyze the data. The $\Delta\Delta Ct$ method was used to analyze the data. Sequences of primers used: β-actin (Forward: 5′-TCGTGCGTGACATTAAGGAG-3′, Reverse: 5′-ATGCCAGGGTACATGGT GGT-3′), TLR1 (Forward: 5′-GCTGATCGTCACCATCGTTG-3′, Reverse: 5′-GTCCACTGGCACACCATCCT-3′) and TLR2 (Forward: 5′-CCTCTCGG TGTCGGAATGTC-3′, Reverse: 5′-GGCCCACATCATTTTCATATACC-3′).

## Flow cytometry

HEK Blue 293 cells co-expressing wild-type or chimera of human TLR1 and TLR2 were seeded at a density of $5 \times 10^5$ cells/ml in 6-well plates. After 24 h incubation, cells were washed with phosphate-buffered saline (PBS) and subsequently digested with 0.25% trypsin (without EDTA). Cells were collected in a 1.5 ml centrifuge tube and washed three times with 1 ml ice-cold PBS. Cells were then stained with FITC anti-TLR1 antibody (Abcam, ab59702) or FITC anti-TLR2 antibody (Abcam, ab59711) at 1/200 dilution for 30 min on ice in dark. Stained cells were washed twice with ice-cold PBS and analyzed by BD Accuri C6 Plus flow cytometry (BD Life Sciences). BD CSample Plus v.1.0 and FlowJo v.10 were used to collect and analyze the data.

## Analysis of the evolutionary conservation of amino acids

To estimate the evolutionary conservation of the amino acids of TLRs we used the ConSurf web server[68], which provides rates of evolution for each amino acid relative to all amino acids in a molecule. The rates were calculated using Bayesian and Maximum Likelihood methods[69]. The calculations were applied to phylogenetic trees that were reconstructed for each TLR separately. The reconstructions were made using the neighbor-joining algorithm as implemented in the ConSurf server based on amino acid multiple sequence alignments (MSAs) for sets of orthologs of each human TLR. The MSAs were taken from the Ensembl genome database project, where they had been precalculated for each human TLR.

## Reporting summary

Further information on research design is available in the Nature Portfolio Reporting Summary linked to this article.

## Data availability

The data that support this study are available from the corresponding authors upon request. The atomic coordinates to the structures in this work have been deposited in the Protein Data Bank (PDB) under accession codes 8AR0 (TLR2tmjm), 8AR1 (TLR3tmjm), 8AR2 (TLR5tmjm) and 8AR3 (TLR9tmjm). The NMR signal assignments have been uploaded in the Biological Magnetic Resonance Bank under codes 34750 [10.13018/BMR34750] (TLR2tmjm), 34751 [10.13018/ BMR34751] (TLR3tmjm), 34752 [10.13018/BMR34752] (TLR5tmjm), 34753 [10.13018/BMR34753] (TLR9tmjm). The source data underlying Figs. 1, 2, 4, 6 and Supplementary Figs. 6, 7, 8, 9, 11, 12, 13, 14, 15, 16, 17, 18, 19 are provided as a Source Data file. Source data are provided with this paper.

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

## Acknowledgements

The work was supported by the Russian Science Foundation grant № 22-14-00020 (K.S.M., structural studies by NMR) and the National Science Foundation of China (No. 21877106 to X.W., No. 22207105 to C.L., functional assays). MD simulations were carried out with the use of computational facilities of the Supercomputer Center "Polytechnical" at the St. Petersburg Polytechnical University and IACP FEB RAS Shared Resource Center "Far Eastern Computing Resource" equipment.

## Author contributions

A.V.S. and M.V.G. synthesized the proteins, F.D.K., V.A.L., and A.V.S. performed the NMR experiments, C.L. did the functional tests, P.E.V. run the computer simulations, E.F.K. investigated the DPPC/DHPC bicelles, A.L.K. synthesized dG-p-dG, A.S.A., S.A.G., X.W., and K.S.M. supervised the work and designed the experiments, K.S.M. and X.W. acquired the funding, K.S.M. and F.D.K. wrote the paper with the assistance of all the authors.

## Competing interests

The authors declare no competing interests
