## [Peer Review File · Nature Communications]

The Architecture of Transmembrane and Cytoplasmic Juxtamembrane regions of Toll-like receptorsReviewers' Comments:

Reviewer #1:

Remarks to the Author:

In this manuscript, four solution-NMR structures of fragments of Toll-like receptors have been determined in detergent micelles or bicelles (for the different proteins, respectively). While the determined structures are significantly different from each other, they have in common that the juxtamembrane domain (JM) is bent to some degree with respect to the trans-membrane (TM) helix. The authors have furthermore designed experiments in which this JM domain is replaced/scrambled/mutated, and investigated the functional consequences (Fig. 6); indeed, these experiments suggest that the JM domain is functionally important.

While I do not have a strong opinion about the biological importance of the study, and do not want to comment on it, I see strengths and weaknesses of this manuscript.

Strengths of the manuscript are:

- * the NMR appears to be state-of-the-art
- * the authors have verified to some extent the effect of the bilayer thickness on the structures (although only indirectly, not by actually determining structures) – see Fig. S12, S13, S14.
- * the structures have been tested *in silico* by MD simulations (although very short ones!)
- * there is some functional relevance to the JM domain, shown by activity measurements (Fig. 6)

The paper also has weaknesses. Among those, the strongest weakness is the choice of the membrane-mimicking media. They are different for the different proteins: TLR2 has been determined in DMPC/DMPG/DHPC 3.2:0.8:10 bicelles, TLR3 and TLR5 have been studied in DPC micelles (a notoriously problematic environment!), and TLR9 in LPPC/LPPG 3:1 micelles.

This leads one to two questions: (i) are the structures representative of what they look like in lipid bilayers? (ii) are the differences between the four structures (see Fig. 3) really reflecting actual differences of these proteins (in a native setting) or are the differences simply due to non-ideal conditions?

In this context, one shall stress that dodecylphosphocholine, which has been used for 2 out of 4 structures, has a long and artefact-filled history in biomolecular NMR. There is even a review on this detergent and how non-native the protein structures in this detergent are: C. Chipot et al., (2018) *Chem. Rev.*, 118 (7), 3559, doi: 10.1021/acs.chemrev.7b00570. To put it bluntly, the authors of that review essentially claim that most alpha-helical structures determined in DPC detergent have distortions, often severe, and often related to altered orientation of helices.

One can indeed see indications that the structures determined here in DPC (TLR3 and TLR5) are, at the very least, unexpected. The beginning and end of the construct is strangely bent back towards where the bilayer would be (Fig 3), and this is exactly for the two structures in DPC. While I do not have a proof, I believe that this is because these parts lie on the detergent micelle surface. In some sense, the current study may be yet another example of structural distortions by DPC.

Note that in the case of TLR3, and to some extent also TLR5, it is physically almost impossible that the beginning and the end of the construct are bent towards the bilayer, namely because in the full-length protein the rest of the protein would simply not allow for this peculiar organization. The rest of the protein would then start in the middle of the bilayer.

Moreover, it is important to realize that these structures do not seem to be stable when put into a lipid bilayer. At least, to me it looks like the structures of TLR3 and TLR5 in Figure 3 (NMR structures) are widely different from those in Figure 5d. This is in fact also the case for TLR9, albeit maybe less dramatic. The differences concern precisely the relative orientations of helices.

In some sense, this latter realization, namely that the NMR structures are not viable in lipid bilayers in MD simulations, shows an inherent inconsistency in the manuscript and raises the question how relevant the structures are.

Note that I do not claim that the NMR structures presented herein are of no use. To test whether one would have gained comparable insight without any experimental data, I have put the sequence of the TLR2 fragment into AlphaFold2, and the prediction shows a straight helix, which is longer than the bilayer. This five-minute test shows that AlphaFold2 is probably not better than this experimental study. Hence, a lot of what the authors report here adds to the knowledge of how these proteins may behave. Nonetheless, I would be very cautious when using the structures for any further use. Along these lines: given how the MD simulation deviates from the starting model (NMR), it would be an interesting test to put an alphafold model into an MD simulation. Maybe the result is similar to those in figure 5, and if so, then the NMR data has not been of much use.

There should be more information about the MD simulations. Given that the protein has diverged quite a bit from the starting state, at least as far as I can tell (Fig 3 vs Fig 5d), I wonder whether this is converged. In fact, there is a statement that makes me think that the structure has not converged and is actually not stable:

"During MD the short N-terminal helices detected in TLR3 joined to the TM helix after the first 50 ns of dynamics. In the case of TLR5 both helices are observed practically till the end of simulations and only on the last frames aggregate into the single helix."

The reader shall have the chance to see the convergence or lack of convergence. This probably means that the trajectories need to be extended.

Overall, I think that the study is well done in a physico-chemical and NMR sense, i.e. the structures are likely very close to what the authors have in their samples. However, I personally am fairly convinced that the structures are not physiologically relevant, possibly with the exception of TLR2 (in bicelles).

I do not want to make statements whether this situation shall or shall not impact the editorial decision to accept this manuscript.

In a revised version, I recommend that the authors address the following points:

The second helix (JM helix) is oriented at about 80° with respect to the bilayer normal, i.e. it seems to lie on the bilayer. If it does so, then it is expected to be amphipathic. Is this the case?

The Results part states "To achieve the most native conditions, we took only the phosphatidylcholine-based lipids and detergents".

I think this statement is very misleading. The detergent that has been used most in this study is dodecylphosphocholine (DPC). This is very well known as a notoriously bad detergent that leads to numerous distortions of protein structure, and it has led to a large number of incorrect structures. See comments above. This should be made very clear in the manuscript.

"TLR3tmjm forms four helical segments" -- Looking at the structure in Figure 3 suggests that there are rather 3, not 4, helical segments. Please clarify.

Figure 5b and c are rather incomprehensible to me. What is meant by "shift" in nanometer? I understand that this is some kind of measure of how much the CA atoms fluctuate, either in the structural ensemble or along the MD trajectory. How exactly has this been calculated?

Figure 1. Please clarify what the color code of the bars means. Is it simply the average hydrophobicity? If so, then the information is somehow doubled; the height of the bars AND the color seem to report the same thing.

In Figure S13, chemical-shift perturbations are calculated. It is not indicated what is the rationale for

applying the factor 1/10 to the ¹⁵N shift changes. There is a lot of literature, e.g. a review by M. P. Williamson *Prog. Nucl. Magn. Reson. Spectrosc.*, vol. 73, pp. 1–16, 2013, doi: 10.1016/j.pnmrs.2013.02.001. I recommend to cite this or other literature to justify the value.

In the methods part, the program PyMol is cited. "Shrodinger" should be "Schrödinger".

The ConSurf software shall be properly cited. Now it is only somewhere in a caption in the supporting information.

The manuscript would benefit from some proofreading by a native speaker; in particular, the articles are not always quite right. This of course does not impact the scientific quality, but there could be some improvements.

Reviewer #2:

Remarks to the Author:

The authors have expressed the intracellular helix regions of few different Toll-like receptors in order to understand the structure and functional roles in signal transduction. NMR structures in the presence of two micelles show that the structures of the intracellular helices and their orientations with respect to the transmembrane helix are indeed dependent on the sequence (families) of TLRs. These are important findings and adds value to our understanding of this fascinating superfamily of TLRs.

I still have a major concern how these regions are likely to behave in the full-length context. For the same reason, I would suggest that the authors downplay the statements at the end (lines 330-335). Have the authors compared the sequences of juxtaposition helices with the eighth helix of GPCRs like rhodopsin? This helix lies perpendicular to the lipid bilayer.

Once these suggestions are considered, the manuscript could be considered further.

Reviewer #3:

Remarks to the Author:

Toll-like receptors are highly studied, evolutionarily conserved signaling molecules that play key roles in pathogen detection. For the most part, TLRs are transmembrane proteins with Leucine-rich extracellular regions (LRRs), transmembrane domains (TMs), and intracellular signaling and, sometimes enzymatic, modules (TIRs). In this manuscript, Korliov et al. have used NMR and computational approaches to model a region near to the internal membrane region which is adjacent to the cytoplasmic domain and they discovered a predicted, conserved, variable length hydrophobic alpha helix that is present in many TLR proteins (including esp TLR2, 3, 5, and 9). This domain appears to be required for signaling to the cytoplasmic domain for downstream activation of TLR-based signaling pathways. Nevertheless, by reconstitution assays in HEK cells, the ordered hydrophobicity of this region seems important for ligand induced signaling to NF-κB for TLR2 (and presumably 3, 5 and 9) but not TLR1 (which lack the domain) receptors.

Overall this appears to be an appropriately performed combination of predictive, biochemical and cell-based experiments, but—to this reviewer---it does not seem to be the type of major advance that meets the level for a major publication.

Specific text comments

--Fig. S18 is confusing. The authors suggest there is a phylogenetic tree, but no such thing seems to be there. Moreover, the molecules that are analyzed are a bit confusing (are they all human ones?).

--The authors make some interesting conjectures about the possible function of this domain, in terms of conformational changes during signaling or adaptor/lipid binding.

RESPONSE TO THE REVIEWERS

First of all, we are very grateful to the reviewers for their expertise and opinion. We received several very constructive comments that, as we hope, will improve our manuscript. To address the comments, we modified the text and figures of the manuscript and ran additional MD simulations. We highlighted all the substantial changes in the text in blue color. Below we provide a point-by-point response to all of the comments.

Reviewer #1:

1. *The paper also has weaknesses. Among those, the strongest weakness is the choice of the membrane-mimicking media. They are different for the different proteins: TLR2 has been determined in DMPC/DMPG/DHPC 3.2:0.8:10 bicelles, TLR3 and TLR5 have been studied in DPC micelles (a notoriously problematic environment!), and TLR9 in LPPC/LPPG 3:1 micelles.*

This leads one to two questions: (i) are the structures representative of what they look like in lipid bilayers? (ii) are the differences between the four structures (see Fig. 3) really reflecting actual differences of these proteins (in a native setting) or are the differences simply due to non-ideal conditions?

In this context, one shall stress that dodecylphosphocholine, which has been used for 2 out of 4 structures, has a long and artefact-filled history in biomolecular NMR. There is even a review on this detergent and how non-native the protein structures in this detergent are: C. Chipot et al., (2018) Chem. Rev., 118 (7), 3559, doi: 10.1021/acs.chemrev.7b00570. To put it bluntly, the authors of that review essentially claim that most alpha-helical structures determined in DPC detergent have distortions, often severe, and often related to altered orientation of helices.

- We agree with the reviewer that DPC and to some extent LPPC are not the most adequate detergents to mimic membranes. In this regard, the fact that three of the final structures were obtained in DPC and LPPC micelles may raise questions. However, we would like to point out, that three out of four proteins (TLR2, TLR5, and TLR9) were actually investigated in DMPC/DMPG/DHPC 3.2:0.8:10 bicelles, despite the fact that the structures of TLR5 and TLR9 were not resolved due to the sample stability and spectrum quality problems. For TLR5 and TLR9 we obtained complete or almost complete NMR chemical shift assignments in all three membrane mimetics that were used in the work: DPC, LPPC/LPPG, and bicelles, which allowed the analysis of the effect of the environment. The data are shown in Figure S11. TLR9 provided almost identical chemical shifts in all three tested mimetics and was stable in MD simulations, which suggests that the obtained structure is likely to be retained in bilayers. In the case of TLR5, we observed the chemical shift differences at the N-terminus, which indicate that the N-terminal bend of the helix may be not retained in bicelles. However, the chemical shifts of the C-terminal helix were also identical in all three mimetics, thus suggesting that the cytoplasmic helix is the element, which does not depend on the type of membrane-like environment. In the case of TLR3, we had only the MD simulations to judge the relevance of the obtained structure. The N-terminal kink was not retained in MD simulations, but the conformation of the cytoplasmic juxtamembrane region was preserved. Thus, we chose to focus not on the differences between the four structures that could be caused by the membrane mimetic but to discuss the similarities - the presence of the cytoplasmic helix that persists in various

membrane mimetics and in MD simulations. We agree that these considerations did not sound or were not properly emphasized in the text of the manuscript. Therefore, we added a separate paragraph to the Discussion section of the revised text, which is devoted to the problems with the relevance of the obtained structures, directly mentions the problems associated with DPC micelles, cites the proposed review, and provides the above-listed evidence.

2. One can indeed see indications that the structures determined here in DPC (TLR3 and TLR5) are, at the very least, unexpected. The beginning and end of the construct is strangely bent back towards where the bilayer would be (Fig 3), and this is exactly for the two structures in DPC. While I do not have a proof, I believe that this is because these parts lie on the detergent micelle surface. In some sense, the current study may be yet another example of structural distortions by DPC. Note that in the case of TLR3, and to some extent also TLR5, it is physically almost impossible that the beginning and the end of the construct are bent towards the bilayer, namely because in the full-length protein the rest of the protein would simply not allow for this peculiar organization. The rest of the protein would then start in the middle of the bilayer.

- In part, we have answered this comment of the reviewer above. We agree that the N-terminal kinks or bends in the transmembrane helices of TLR3 and TLR5 may be the result of the detergent influence, and for the case of TLR3, we explicitly state it in the Discussion section of the manuscript. However, the cytoplasmic juxtamembrane part structure is retained in all the mimetics for TLR5 and in the MD simulations of TLR3, thus we suggest that these parts of the obtained structures actually describe or are close to the conformations of the domains in the lipid bilayer. We also agree that the N-terminal kink of TLR3 would be incompatible with the presence of the ligand-binding domain and now state it in the revised version of the manuscript. On the other hand, the C-terminal TIR domains are connected to the juxtamembrane helices by quite long flexible linkers of 12-14 amino acids, which could easily adapt to the observed mutual arrangement of juxtamembrane and transmembrane helices.

3. Note that I do not claim that the NMR structures presented herein are of no use. To test whether one would have gained comparable insight without any experimental data, I have put the sequence of the TLR2 fragment into AlphaFold2, and the prediction shows a straight helix, which is longer than the bilayer. This five-minute test shows that AlphaFold2 is probably not better than this experimental study. Hence, a lot of what the authors report here adds to the knowledge of how these proteins may behave. Nonetheless, I would be very cautious when using the structures for any further use.

Along these lines: given how the MD simulation deviates from the starting model (NMR), it would be an interesting test to put an alphafold model into an MD simulation. Maybe the result is similar to those in figure 5, and if so, then the NMR data has not been of much use.

- In our opinion, AlphaFold2 is a "black box" tool, which provides predictions (sometimes very good, sometimes not) with unknown reliability. AlphaFold2 would not predict the effect of lipid composition and specific protein-lipid interaction on the protein structure and

both are extremely important, taking into account the mosaic nature of cell membrane. Thus, we suggest that experimental data, obtained in a variety of membrane-like conditions, are valuable, regardless of the AlphaFold2 prediction results.

4. There should be more information about the MD simulations. Given that the protein has diverged quite a bit from the starting state, at least as far as I can tell (Fig 3 vs Fig 5d), I wonder whether this is converged. In fact, there is a statement that makes me think that the structure has not converged and is actually not stable:

"During MD the short N-terminal helices detected in TLR3 joined to the TM helix after the first 50 ns of dynamics. In the case of TLR5 both helices are observed practically till the end of simulations and only on the last frames aggregate into the single helix."

The reader shall have the chance to see the convergence or lack of convergence. This probably means that the trajectories need to be extended.

- To take into account this comment of the reviewer, we calculated additional 500 ns of MD simulation, the results are shown in the revised version of the manuscript, Figure 5. No additional events took place for TLR2, TLR5, and TLR9, while in the case of TLR3 the region that was initially an N-terminal short helix and then joined the TM domain, became partially destabilized in the last 200-300 ns.

5. *The second helix (JM helix) is oriented at about 80° with respect to the bilayer normal, i.e. it seems to lie on the bilayer. If it does so, then it is expected to be amphipathic. Is this the case?*

- Indeed, the JM helices of TLR2 and TLR9 not only lie on the bilayer surface but are immersed into the hydrophobic interior of the bilayer, to allow direct contact between the juxtamembrane residues and lipid methylene groups (Figure 4). To answer this comment of the reviewer, we analyzed the sequence of the juxtamembrane helix using two scales: the White and Wimley octanol scale, which describes the propensity of residues to be in the hydrophobic interior of the membrane; and the POPC interface scale, which describes the propensity of residues to enter the lipid/protein interface. The analysis reveals that juxtamembrane helices are not amphipathic - there is no pronounced asymmetry between the polar and hydrophobic residues. On the other hand, the juxtamembrane domains are extremely rich with amino acids that favor the water/lipid interface, which explains the surface-associated position of the helices. The classical analysis that was suggested by Wimley and White (averaging in the frame of 15-20 amino acids) clearly reveals the TM region, with a high score in the octanol-interface scale and a juxtamembrane region with a high score in the interface scale and low score in the octanol-interface scale (Figure R1). We added this analysis to the revised manuscript and now provide its results in the Discussion section, the results are shown in the new figure S18.

Residue

Figure R1. Whole-residue hydropathy plots for TLRs, for which spatial structures of transmembrane and cytoplasmic juxtamembrane regions were determined. For each amino acid, the average hydrophobicity (the frame equals fifteen amino acids) according to White-Wimley scale was calculated that shows the free energies of transferring amino acids from water to POPC interface (POPC interface), from water to n-octanol (octanol) and difference between these two scales (octanol - POPC interface). Yellow and cyan backgrounds indicate transmembrane (TM-helix) and juxtamembrane (JM-helix) helices according to this work and previous work on TLR4.

6. *The Results part states "To achieve the most native conditions, we took only the phosphatidylcholine-based lipids and detergents". I think this statement is very misleading. The detergent that has been used most in this study is dodecylphosphocholine (DPC). This is very well known as a notoriously bad detergent that leads to numerous distortions of protein structure, and it has lead to a large number of incorrect structures. See comments above. This should be made very clear in the manuscript.*

- In the revised version, we state it directly in the Results section and again in the new paragraph of Discussion. In the Results section, we state: "**DPC is known to cause the structure distortion in helical membrane proteins [10.1021/acs.chemrev.7b00570], therefore it was considered only if all other mimetics failed to provide the necessary spectrum quality and sample stability.**"

7. *"TLR3tmjm forms four helical segments" -- Looking at the structure in Figure 3 suggests that there are rather 3, not 4, helical segments. Please clarify.*

- According to the structure analysis (Figure S10). There was a bend at the C terminus of the transmembrane helix of TLR3. We agree that the bend is rather subtle, and changed the description of the TLR3 structure. Now we state that there are three helical segments.

8. *Figure 5b and c are rather incomprehensible to me. What is meant by "shift" in nanometers? I understand that this is some kind of measure of how much the CA atoms fluctuate, either in the structural ensemble or along the MD trajectory. How exactly has this been calculated?*

- The shift in nm is the standard deviation of distances between the CA atoms of the corresponding residues, calculated for the set of the structures (either the NMR set, or the structures along the MD trajectories). The resulting maps reveal the parts of the protein that move synchronously and the parts that move independently from each other (domains or subdomains). We added the explanatory sentences to the Results section and to the Figure 5 caption, to make it clear.

9. *Figure 1. Please clarify what the color code of the bars means. Is it simply the average hydrophobicity? If so, then the information is somehow doubled; the height of the bars AND the color seem to report the same thing.*

-Indeed, the bars are colored with respect to the average hydrophobicity, to make the plot more readable. We added the note to the figure legend.

10. In Figure S13, chemical-shift perturbations are calculated. It is not indicated what is the rationale for applying the factor 1/10 to the 15N shift changes. There is a lot of literature, e.g. a review by M. P. Williamson *Prog. Nucl. Magn. Reson. Spectrosc.*, vol. 73, pp. 1–16, 2013, doi: 10.1016/j.pnmrs.2013.02.001. I recommend to cite this or other literature to justify the value.

- We added the rationale for choosing factor 1/10 and cited the proposed review in the Methods section of the revised manuscript.

11. In the methods part, the program PyMol is cited. “Shrodinger” should be “Schrödinger”.

- We are grateful for this remark and corrected the citation in the revised version of the manuscript.

12. The ConSurf software shall be properly cited. Now it is only somewhere in a caption in the supporting information.

- This was our mistake to not describe this part of the work in the experimental part of the manuscript. We added a more detailed description of how we analyzed the evolutionary conservation of amino acids of TLRs to the new paragraph of the Methods section of the revised manuscript and named it **Analysis of the evolutionary conservation of amino acids** and cited works related to the ConSurf Server appropriately.

13. The manuscript would benefit from some proofreading by a native speaker; in particular, the articles are not always quite right.

- The revised manuscript underwent thorough editing with the help of a language expert.

Reviewer #2:

1. I still have a major concern how these regions are likely to behave in the full-length context. For the same reason, I would suggest that the authors downplay the statements at the end (lines 330-335).

- We agree with the reviewer, our structures were obtained in the truncated protein constructs and in membrane mimetics, which could affect the obtained structures, we now acknowledge it clearly in the Discussion of the revised manuscript. However, we would like to point out that this refers to almost all the structural data available for TLRs. Extracellular and intracellular domains are known to fold independently, therefore one can expect the same for the membrane-associated part of the protein. Additionally, we would like to note that we investigated our proteins in a variety of membrane mimetics and *in silico* in lipid bilayers, and in all the cases, the presence of cytoplasmic juxtamembrane membrane-associated helix was retained. According to the request of the reviewer, we modified the final paragraph of the manuscript. Now we state: *"Together with the*

previously published structure of TLR4³⁰, we can state that for all five TLR subfamilies, conformations of the membrane-associated parts of receptors are now available, however, in a membrane mimetic environment and within the truncated protein constructs."

2. Have the authors compared the sequences of juxtaposition helices with the eighth helix of GPCRs like rhodopsin? This helix lies perpendicular to the lipid bilayer.

- To answer this comment, we analyzed the sequence of the 8th helix in GPCR-like rhodopsin. According to our data, helix 8 has little in common with the JM helix in TLRs. Helix 8 is highly polar, with only 3-4 hydrophobic residues, while the JM helices are extremely hydrophobic and contain almost no polar side chains. In addition, helix 8 is palmitoylated at the C-terminus, and it is, most likely, the major reason why this part of the protein forms a surface-associated helix. In turn, the JM helices of TLRs are rich with aromatic sidechains (Trp, Tyr), which favor the surface-associated arrangement of the helix.

Reviewer #3:

1. Fig. S18 is confusing. The authors suggest there is a phylogenetic tree, but no such thing seems to be there. Moreover, the molecules that are analyzed are a bit confusing (are they all human ones?).

- This was our mistake to not describe this part of the work in the experimental part of the manuscript. We added a more detailed description of how we analyzed the evolutionary conservation of amino acids of TLRs to the new paragraph of the Methods section of the revised manuscript and named it **Analysis of the evolutionary conservation of amino acids**. One of the intermediate steps of calculating a conservative score was a reconstruction of a phylogenetic tree. This step was done in Consurf Server, where one could upload a multiple sequence alignment (MSA) to reconstruct a phylogenetic tree. We decided to perform an analysis on MSAs of TLRs orthologues. In the Ensemble genome database project (<https://www.ensembl.org/index.html>) we found genes for each human TLRs. In this database, the sets of orthologs for each TLR are already found among all genomes that are present in the database. So, one could download an MSA of orthologs for a particular TLR.

Reviewers' Comments:

Reviewer #1:

Remarks to the Author:

The authors have, in my opinion, addressed the concerns I have raised. Clearly, this study represents quite some work, and the authors have tried to perform cross-checks (several membrane-mimicking environments, MD). I am not fully convinced that all structures are relevant and even physically possible in a cell. But the reader can now better make their own opinion. Technically, the study is well done, at least besides the issue of the membrane-mimicking environment (DPC in particular).

Reviewer #2:

Remarks to the Author:

I am happy with the current version of the manuscript and I recommend acceptance of the same.

Reviewer #3:

Remarks to the Author:

Toll-like receptors are highly studied, evolutionarily conserved signaling molecules that play key roles in pathogen detection. In this revised manuscript, Korliov et al. have used NMR and computational approaches to model a region near to the internal membrane region which is adjacent to the cytoplasmic domain and they discovered a predicted, conserved, variable length hydrophobic alpha helix that is present in many TLR proteins (including esp TLR2, 3, 5, and 9). This domain appears to be required for signaling to the cytoplasmic domain for downstream activation of TLR-based signaling pathways. By reconstitution assays in HEK cells, the ordered hydrophobicity of this region seems important for ligand induced signaling to NF- κ B for TLR2 (and presumably 3, 5 and 9) but not TLR1 (which lack the domain) receptors.

Overall, the authors have performed a combination of predictive, biochemical and cell-based experiments, and modified and toned back the presentation of (esp structural) data and conclusions in this paper. Moreover, they have appropriately changed the sequence comparison data in light of my previous comment. Still, the overall conclusion is rather incremental for TLR biology, and there is still a lack of information of how this domain functions for signaling.

RESPONSE TO THE REVIEWERS

We would like to thank all reviewers for their thorough analysis of our work that helped us to improve the article and overall positive opinion.

Reviewer #1:

1. *The authors have, in my opinion, addressed the concerns I have raised. Clearly, this study represents quite some work, and the authors have tried to perform cross-checks (several membrane-mimicking environments, MD). I am not fully convinced that all structures are relevant and even physically possible in a cell. But the reader can now better make their own opinion. Technically, the study is well done, at least besides the issue of the membrane-mimicking environment (DPC in particular).*

- We suppose that the question of relevance would persist regardless of the mimetic systems used in the work. Even the liposomes are quite far from the real cell membranes. We believe that with all the cross-checks our data provides an insight into the architecture of membrane-associated parts of TLRs and shows the way how these parts adapt to the changes of the lipid environment.

Reviewer #2:

1. *I am happy with the current version of the manuscript and I recommend acceptance of the same.*

- We thank the reviewer for his/her positive assessment of our work.

Reviewer #3:

1. *Overall, the authors have performed a combination of predictive, biochemical and cell-based experiments, and modified and toned back the presentation of (esp structural) data and conclusions in this paper. Moreover, they have appropriately changed the sequence comparison data in light of my previous comment. Still, the overall conclusion is rather incremental for TLR biology, and there is still a lack of information of how this domain functions for signaling.*

- We agree with the reviewer that the detailed mechanism of TLRs activation remains to be elucidated and we believe that our data provide the base to us and other groups for the planning of new experiments and thus make the solution closer.